# Perceiving the Role of Communication Skills as a Bridge between the Perception of Spiritual Care and Acceptance of Evidence-Based Nursing Practice—Empirical Model

**DOI:** 10.3390/ijerph182312591

**Published:** 2021-11-29

**Authors:** Mariusz Panczyk, Lucyna Iwanow, Szymon Musik, Dominik Wawrzuta, Joanna Gotlib, Mariusz Jaworski

**Affiliations:** Department of Education and Research of Health Sciences, Faculty of Health Sciences, Medical University of Warsaw, Zwirki i Wigury 81, 02091 Warsaw, Poland; lucyna.iwanow@wum.edu.pl (L.I.); szymonmusik13@gmail.com (S.M.); dominik.wawrzuta@wum.edu.pl (D.W.); joanna.gotlib@wum.edu.pl (J.G.); mariusz.jaworski@wum.edu.pl (M.J.)

**Keywords:** evidence-based practice, spiritual care, communication skills, path analysis

## Abstract

Decision making using evidence-based practice (EBP) is generally universally accepted by nurses. Such acceptance may affect the personnel’s behaviour towards patients, which is also demonstrated by taking into consideration the patient’s preferences, including the patient’s spiritual needs, in the care plan. The provision of such care requires the development of an attitude of approval and an adequate level of communicative competence, which will enable the actual implementation of the EBP. The purpose of our study was to assess the perception of spirituality and the nurse’s role in providing spiritual care, as well as the perception of the significance of communication skills in the approval of EBP in professional practice. A multi-centre cross-section study was conducted on a population of 1176 participants (459 undergraduate (bachelor’s programme, BP) and 717 postgraduate students (master’s programme, MP)) from 10 medical universities in Poland. Three tools were used in the study to evaluate the participants’ approach: Evidence-Based Practice Competence Questionnaire (EBP-COQ), The Spirituality and Spiritual Care Rating Scale (SSCRS), and Communication Skills Attitude Scale (CSAS). Structural equation modelling was used for the analysis. An analysis of structural equations revealed the presence of positive relationships of the attitude to spiritual care and the role of communicative competences with the approach to EBP regardless of the cohort. A significant difference was found related to the influence of age on the attitude towards learning communicative competences. The approval in this respect was observed to decrease with age in the MP group. Increasing approval of EBP requires strengthening the approach to activity-centred spiritual care, with the simultaneous development of a positive attitude towards learning communicative competences. The model reveals the need to integrate a humanistic approach with EBP, which can be achieved by planning different interventions in different groups of recipients: nurses, academic teachers and students.

## 1. Introduction

According to the International Council of Nurses, evidence-based practice (EBP) in nursing is defined as “a problem solving approach to clinical decision making that incorporates a search for the best and latest evidence, clinical expertise and assessment, and patient preference values within a context of caring” [1]. It can be stated that EBP is based on decision making and is used to optimise patient outcomes, improve clinical practice, and ensure accountability in nursing [2]. The concept of EBP is characterised by a high level of acceptance by nursing students and working nurses [3], but there are still real barriers to its implementation in clinical practice [4].

As mentioned before, patient preference values and expectations are among the three EBP components that are most essential to therapeutic success. The roles of all three components are considered equal: scientific evidence, clinical expertise and assessment [5]. Good nursing care in EBP involves scientific evidence and knowledge derived from best practice being balanced with knowledge of the values and preferences of the individual patient [6].

The first nursing models emphasised the patient’s needs and expectations as an essential component of care. The model developed by Virginia Henderson, based strictly on Maslow’s theory of needs, accentuates the therapeutic and caring role of the nurse, and refers to the patient’s essential needs that the nurse is supposed to satisfy. This model applies not only to biological needs, but also to such higher needs as spirituality or contact with other humans [7]. The spiritual and humanistic dimension of care initiated by Henderson became the core of Jean Watson’s model, which focused on the concepts of care and concern [8].

The important role of the nurse’s communicative competence in making clinical decisions related to the EBP concept was also recognised and described in Hildegard Peplau’s model [9]. It is not possible to understand the patient and identify their needs and expectations related to the therapeutic process without an excellent patient–nurse relation. Such a good relationship requires the nurse’s positive approach (positive attitude) to learning communication skills. In the context of psychology, attitudes are a set of different variables (e.g., cognitive, emotional, etc.) that affect one’s behaviour. Moreover, there is a strong relationship between the attitude and the behaviour. In other words, a change in the attitude entails a change in the behaviour [10].

A decision related to the direction of therapeutic and care activities can be made only when the professional’s (nurse, physician) mindset is taken into consideration; however, it can also partly or fully consider the patient’s opinions and wishes [5]. The active participation of the patient entails the need to include in the care situation their preferences, such as personal norms, values, characteristics, and wishes. They often also include spiritual needs, which the nurse must take care of to satisfy the patient’s emotional needs. The provision of spiritual care by a nurse means that the nurse must employ a wide range of communication tools, without which the care cannot be provided effectively.

There are several publications devoted to nurses’ perceptions of the role of communication skills in the context of EBP [5,11] and spiritual care [12]. The number of papers analysing the nurses’ attitude towards these skills is limited [13]. More importantly, there are no studies analysing these issues together, as three interrelated aspects of nursing care, in one integrated model.

### 1.1. Conceptual Framework

The proposed theoretical model assumes that the provision of activity-centred spiritual care (ACSC) will be positively related to the attitude towards EBP (Figure 1). The positive attitude towards learning communicative competences has been assumed to play an intermediate role. This assumption is based on the observations made by Wittenberg et al. (2017) [12], who indicated that an adequate level of communication skills determines the correct and effective provision of spiritual nursing care. This applies in particular to nonverbal communication, listening, and discussing patients’ emotions. Concerning the above, the nurses who want to provide spiritual care effectively tend to aware of the need to improve their communication skills, which can translate onto a positive attitude towards learning communicative competences (Hypothesis 1). A positive attitude towards learning communicative competences should, in turn, lead to a positive attitude towards EBP, which is suggested, e.g., in the studies conducted by Adams et al. (2017) [11], Allenbaugh et al. (2019) [14] (Hypothesis 2). The age of the nurse can also be of key significance in the proposed model (Hypothesis 3 and 4). The available test results do not allow for stating clearly the extent to which the age or job seniority can condition a positive attitude towards learning communicative competences or the provision of activity-centred spiritual care [15].

### 1.2. Aim

To assess the perception of spirituality and the nurse’s role in providing spiritual care, as well as the perception of the significance of communication skills in the approval of EBP in the professional practice.

## 2. Methods

### 2.1. Design

A multi-centre cross-sectional national study was conducted in Poland from February to June 2019.

### 2.2. Sample and Setting

Nine Polish medical universities educating students in first-cycle (corresponding to the 6th Level of the European Qualifications Framework) and second-cycle studies (corresponding to the 7th Level of the European Qualifications Framework) of a nursing major were invited to participate in the study. Seven universities agreed to participate in the study and conducted data collection in student groups. These seven universities educate 30% and 50% of all Polish students undertaking a major in nursing in both cycles of study, respectively.

The study population consisted of two subgroups: 459 undergraduate (bachelor’s programme, BP) and 717 postgraduate students (master’s programme, MP). For the BP group, being a student in the last (third) year of studies was the inclusion criterion. For the MP group, being a first- or second-year student and declaring at least three years of professional experience were the inclusion criteria.

A total of 1475 students were qualified for the study, and necessary data were obtained from 1176 of them (response rate 79.7%). With this sample size and the number of nursing students in Poland (*n* = 9000), the error margin was 2.72% (95% confidence level).

### 2.3. Ethical Considerations

Before taking part in the study, voluntary informed consent to participate was obtained from each participant. The aim of the study, as well as the methods of analysis and data storage, was explained to the participants in writing. The participants were also informed that confidential data would be used for scientific purposes only. The Local Personal Data Inspector raised no contraindications regarding the data protection of the study participants.

### 2.4. Variables

We used the Evidence-Based Practice Competence Questionnaire (EBP-COQ) created by Ruzafa-Martinez et al. (2013) [16], in the form of the Polish adaptation constructed by Panczyk et al. (2020) [17]. The questionnaire consists of 25 items arranged into three subscales: attitude towards EBP, skills in EBP, knowledge in EBP. The scoring range obtained by measuring with the EBP-COQ is 25 to 125. The EBP-COQ had good reliability as calculated during the validation study (Cronbach’s alpha = 0.856). As this publication is focused on the fundamentals, the authors only used the scale referring to the attitude towards EBP. This subscale has good reliability, which was alpha = 0.798.

The Spirituality and Spiritual Care Rating Scale (SSCRS) developed by McSherry et al. (2002) [18] is used for the measurement of spirituality and spiritual care in nursing. The scale consists of 17 items, with subscales emotional support-centred spiritual care, ACSC, and religiosity. The scoring range obtained by measuring with the SSCRS is 17 to 85. The 17-item SSCRS used in the study demonstrated a reasonable level of internal consistency reliability, a Cronbach’s alpha coefficient of 0.640. The authors of the paper analysed only the ACSC subscale, the reliability of which was alpha = 0.894.

The Communication Skills Attitude Scale (CSAS) constructed by Rees et al. (2002) [19] (Polish adaptation by Panczyk et al. (2019) [20]) was used for the measurement of attitudes towards learning communicative competences. The validated Polish version contains 23 statements pertaining to the attitude towards learning communicative competences in both teaching the profession and in professional practice. The scoring range obtained by measuring with the CSAS is 23 to 115. The scale is divided into two subscales: a positive one and a negative one. In a validation study, both subscales showed good internal consistency (Cronbach’s alpha 0.901 and 0.802, respectively). The final CSAS result was used in the study.

Some sociodemographic data of the study group were collected, including the nursing school, age, and job seniority, as well as previous participation in training for EBP, communication skills or spiritual care.

### 2.5. Data Collection

The data were collected by means of an auditorium survey. Students sitting in one room after their class were asked to fill in the questionnaire. The trained interviewers communicated the purpose of the study orally and informed the participants of how to fill in the questionnaires. The interviewers were also responsible for collecting the filled-in questionnaires and securing them before they were sent to the central unit coordinating the study.

### 2.6. Data Analysis

Descriptive statistics methods (means and standard deviations) and structure indicators (numbers and frequencies) were used for the analysis of the variables collected in the study. In order to compare the two study groups (BP vs. MP) regarding the relevant characteristics, a chi-square independence test or Student’s t test was used, depending on the type of the variable. The calculations were performed using the STATISTICA package, version 13.3 (Tibco Software Inc., Palo Alto, CA, USA). A 0.05 significance level was set.

All analyses were carried out using the structural equation modelling program Mplus version 7.0 [21]. We used two-group structural equation modelling: BP vs. MP. The purpose of the analysis was to obtain an answer to the question of whether the collected empirical data confirm the relationship between the variables that was assumed by the researchers. To that end, the model’s parameters were estimated (path coefficients, variance and covariance) and these were used for building a theoretical variance–covariance matrix of the variables employed in the model (Figure 2). We checked whether the calculated model’s parameters varied for BP vs. MP. Maximum likelihood estimation with robust standard errors was used to calculate the structural model’s parameters.

The fit of the model was assessed using the following statistics and indices: the chi-square test of model fit (CMIN), normal chi-square (CMIN/DF), the Comparative Fit Index (CFI), the Tucker–Lewis Index (TLI), the root mean square error of approximation (RMSEA), and standardised root mean square residual (SRMR). In the evaluation of the model, the chi-square statistics were expected to be insignificant. The recommended values of indices were as follows: χ2 divided by the degrees of freedom (CMIN/DF) ≤ 3.00; RMSEA < 0.080 and SRMR < 0.050; CFI and TLI > 0.95 [22].

## 3. Results

### 3.1. Participant Characteristics

Both groups of study participants varied in a statistically significant way by the mean age (BP vs. MP: 23.1 (3.79) vs. 32.6 (10.91); *t* = 17.861, *p* < 0.001). Regarding the selected characteristics, the groups varied significantly for such aspects as nursing school, place of residence, and spiritual care training. Table 1 reports a comparison of the selected demographic characteristics of the participants who completed this study.

### 3.2. Variables

Skewness and kurtosis were evaluated for all the analysed data, which revealed a left-sided asymmetry and a lack of compliance with a normal distribution. The deviations in compliance with normal distribution were not very high because the skewness and kurtosis, which ranged from −1.5 to +1.5.

We also analysed whether the two study groups varied significantly with regard to the tested features. The mean intensity of the characteristics was observed to be lower in a statistically significant way in the BP than the MP group. The effect sizes were not similar between these variables. The details of the results are summarised in Table 2.

### 3.3. Measurement Model

Both calculated values (CMIN = 1.797, df = 4 and CMIN/DF = 0.45) indicate that the assumed theoretical model is confirmed by the empirical data. The CMIN contribution from each group amounted to 0.529 (CMIN/DF = 0.12) for BP and 1.268 (CMIN/DF = 0.32) for MP. Moreover, based on the obtained value of the test probability (*p* = 0.773), the hypothesis regarding the lack of difference between the theoretical and empirical variance–covariance matrixes was highly likely correct. The results of the test invariance across groups indicate that the model varies in a statistically significant way between the two study groups (CMIN = 10.863, df = 3, *p* = 0.012) for the path parameters.

The value of the divergence function F0 and the value of the RMSEA index corrected by the number of the degrees of freedom were determined. The value of RMSEA (0.001, 90%CI (0.000; 0.042), *p* = 0.975) suggests that both matrices are equal. The value of SRMR was below the assumed threshold and amounted to 0.009. The results confirm a good fit of the data to the assumed structure of the model.

In order to more accurately estimate the degree of model fit to the collected data, the fit indices were determined, and we compared the tested model with the depended model (the model in which the variables are not correlated). The CFI value (0.999) and its corrected value, i.e., TLI (1.000), were estimated. The results suggest that the tested model explains nearly 100% of the variable variance.

### 3.4. Associations between Variables

An analysis of the proposed path model revealed that all relations had a positive effect. The CSAS variable was observed to have a significant direct effect on the explained variable (EBP). For this relation in the BP and MP groups, the values of the standardised regression weights amounted to 0.327 (*p* < 0.001) and 0.332 (*p* < 0.001), respectively. The observed relationship did not vary in a statistically significant way between BP and MP (CMIN = 0.086, df = 1, *p* = 0.769).

The effects of the two variables (Age and ACSC) on CSAS were different in the two compared groups. Although in the BP group, the ACSC variable exerted a fairly strong effect on CSAS, in the MP group, the relationship was significantly weaker. The values of the standardised regression weights amounted to 0.347 (*p* < 0.001) and 0.160 (*p* < 0.001), respectively, and the difference between BP and MP was statistically significant (CMIN = 9.643, df = 1, *p* = 0.002). In reference to the effect of age, it was observed that in the BP group, the variable did not affect CSAS in a significant way (0.037, *p* = 0.398), whereas such an influence was observed in the MP group (0.255, *p* < 0.001).

As far as the correlation between Age and ACSC is concerned, no statistically significant relationship was observed in either the BP or the MP group (*p* value 0.834 and 0.057, respectively). The details of the parameter estimation used for a model of structural equations, including a comparative analysis of the groups, are shown in Table 3.

By analysing the model and the direct effects, we determined the value of indirect effects (where the effect of the explaining variables on the explained variable is not direct). The first of the indirect effects in the presented model was executed according to the ACSC → CSAS → EBP path (standardised regression weight in BP and MP group: 0.113 and 0.053, respectively). This indirect effect was much weaker in the BP than in the MP group. The other indirect effect was executed via the Age → CSAS → EBP path (standardised regression weight in the BP and MP group: 0.012 and 0.085, respectively). This indirect effect was thus much weaker in the BP group. The details of the parameter estimation for a model of structural equations, with all direct and indirect effects, are shown in Figure 3.

## 4. Discussion

The role of the attitude towards learning communicative competences, as a bridge between the perception of spiritual care and the approval of EBP, was analysed based on a previously developed theoretical model, which was verified empirically. The theoretically assumed relationships between the analysed variables (attitude towards learning communicative competences, ACSC, attitude towards EBP) were confirmed by the tested model in two groups: BP and MP. Given that nursing practice is based on team collaboration, the comparison of the groups was based on the assumptions of management theories, which emphasise the role of age diversity and experience in the development of a friendly working environment [23,24].

The possibility of comparing the results for the path model based on the data collected in the BP and MP groups allows to observe some significant differences that can be determined by the heterogeneity of both groups. There are students in the BP group who are still in the process of shaping their attitudes and opinion (also abut EBP and spiritual care). Along with learning about new fields of nursing and practice in clinical classes, attitudes that will determine future professional behaviour are formed. Contrary to the participants from the MP group who is a due to their previous professional experience, may perceive nursing practice differently (including spiritual care and EBP). The analysed path model shows some differences in both tested groups (BP vs. MP).

In the context of the contemporary management of nursing staff, a positive attitude towards learning communication skills as well as proceeding according to EBP both play an important role, which is a real challenge in everyday clinical practice. This may be related to the existence of different educational paths in the area of EBP, or the lack of such paths in the curricula, as well as several constraints that hamper the implementation of EBP in organisations [4]. Currently, two different approaches to nursing practice may be employed in one team—one based on scientific evidence, and one based on personal convictions and intuition resulting from experience [25,26]. The two different attitudes may affect the nurses’ approach to learning communication skills and using them in nursing care. By observing the behaviour of other people in their environment, a nurse can follow a behavioural pattern as a result of behaviour modelling. This is of particular importance in reference to nurses starting their professional career right after graduation. Their idealistic approach can be modified when they face the reality of clinical practice and observe the behaviours of their colleagues [27,28].

The collected data reveal that the nurse’s age and related life experience affect their attitude towards the variables studied in the developed model. The age of nurses was directly positively correlated with the attitude towards learning communicative competences and was indirectly related to the attitude towards EBP. Hence, the attitude towards learning communicative competences can depend on whether the person has already started their career or is still learning, supervised by a teacher under simulated conditions, which are often idealised as compared to the reality of working in hospital wards. These deliberations can be confirmed by personal job experience and discussions with experienced nurses, as well as with persons who have just graduated from their own nursing studies. These observations reveal that not all graduates or active nurses are satisfied with the level of their communication skills, and more importantly, they express the need to strengthen them. A similar opinion can be found in studies on nurses’ soft skills [29], studies devoted to patients’ needs [14] and the theoretical model of the nursing theory developed by Watson and Smith [8].

Watson emphasises that communication skills should be highly developed due to the specificity of a nurse’s job [8]. The model proposed in the study strongly corresponds with Watson’s humanistic theory, which places human needs in the central position. The patient’s needs can be satisfied by building a therapeutic bond, which requires adequate communication skills from the nursing staff [30]. Furthermore, adequate communication skills are considered to contribute directly to the reduction in medical errors and the increase in patient safety, as well as to making the care plan better and more efficient, which means shorter hospitalisation time for the patient [24]. This is also reflected in the developed model, wherein regardless of the study group (BP or MP), the attitude towards learning communicative competences is positively correlated with all the variables. Studies by Mattson et al. (2015) [24] and Kirca and Bademli (2019) [29] have revealed that the development of communication skills brings several benefits—both for the staff’s development and for competence strengthening— as they contribute to professional adaptation and reduce the risk of burnout.

Another important aspect of the study that differentiates the two analysed groups is the perception of spiritual care. The developed model suggests that professional experience can affect the perception of the nurse’s role in providing spiritual care, which is among the most essential areas of nursing care and is based both on adequate communication skills [8] and a specific mental disposition (e.g., empathy) and self-awareness. Physical confrontation with the patient’s spiritual needs requires specific measures to be taken by the nurse, which should comply with the adopted standards and EBP. This, in turn, increases self-awareness in advancing one’s skills in the area of spiritual care according to the EBP concept, which includes the patient’s expectations, preferences and individual needs. In other words, professional experience can modify the idealised perception of one’s own skills in providing spiritual care, and a shift from “intuitive proceeding” to more evidence-based proceeding. In spiritual care, it is crucial to understand another person’s situation without judging them and to remain open to support them in times of hardship. It thus means referring to facts rather than to one’s emotions or convictions.

The presented model demonstrates an indirect relationship between the perception of ACSC and the attitude towards EBP. The attitude towards learning communicative competences is the bridge between the variables. It can be assumed that a positive approach to learning communication skills affects a nurse’s self-assessment and his/her sense of being capable of entering into difficult discussions with the patient, as well as with members of the nursing team or other health care professionals [31]. Hence, communication skills and spiritual care are strongly related to EBP because each of them strives towards the patient’s conscious participation in the therapeutic process. In this context, the nurse’s approach to the integrated (simultaneous) learning of competences is essential.

The developed model not only indicates the possibility of integrating a humanistic approach with EBP, but it also enables the conscious and insightful influence of different groups of recipients: nurses, managers of the health care system or of the nursing staff, academic teachers, and nursing students.

Knowledge of the relationships between the attitude towards learning communicative competences, ACSC, and the attitude towards EBP can help nurses to make more conscious decisions regarding undertaking postgraduate education. It applies both to education and self-improvement, which takes into consideration the balance between hard and soft competencies. It can significantly strengthen one’s sense of self-confidence in making independent decisions based on EBP, by incorporating a humanistic aspect into the nursing practice. The approach presented in the model will help to improve the quality of care, reduce the risk of burnout, and enable easier adaptation to the working environment.

Academic teachers can take advantage of the observations resulting from the model for the better execution of practical classes based on EBP, and for developing a curriculum covering the simultaneous teaching of hard and soft competencies. It will help improve teaching effectiveness. Consequently, staff with an adequate scope of competences will be able to provide the highest level of care [32].

It should be emphasised that the developed empirical model integrates EBP with a humanistic approach, encouraging the perception of communication skills as a stable and robust bridge between the two areas. To a certain extent, the model varies depending on whether it applies to working professionals or students, but the pillars of the concept remain the same. Only the emphasis is shifted between the pillars. The presented model highlights the importance of self-awareness and self-diagnosis of one’s strengths and weaknesses. This fits the current concept of nurses’ professional independence.

### Study Limitations

The interpretation of the results is subjected to some constraints that must be taken into consideration when evaluating the results. First and foremost, working nurses who decided to undertake postgraduate studies were included in the study. Consequently, this group may be more aware of the need to improve their skills. The study did not include working nurses who did not continue their postgraduate education. Declarative participation in different types of training (e.g., spiritual care training, EBP training and communication skills training) was controlled in the study, without our having learnt the thematic scope of the training. The model was developed based on data from a cross-sectional study.

## 5. Conclusions

The developed empirical model integrates the concept of EBP with the humanistic component of emphasising the patient’s needs, which should become the centre of the nurse’s attention. The practical execution of this kind of nursing care requires well-developed communicative competences, which create a stable bridge linking the two different approaches. The relations comprising the present model include not only a holistic approach to nursing care, but also guidelines regarding practical actions that can enable conscious and insightful educational actions (e.g., among students, nurses and academic teachers).

## Figures and Tables

**Figure 1 ijerph-18-12591-f001:**
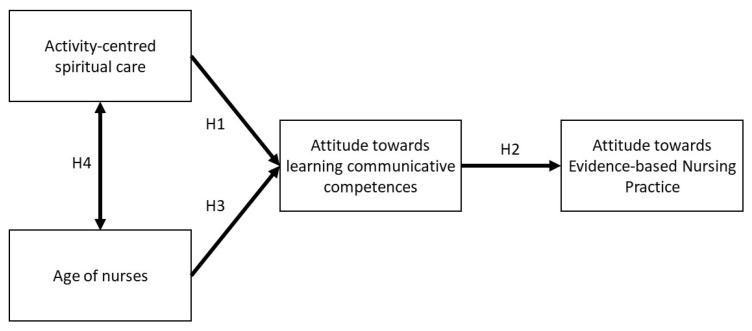
The theoretical model of the relationship between the tested variables (H—hypothesis).

**Figure 2 ijerph-18-12591-f002:**
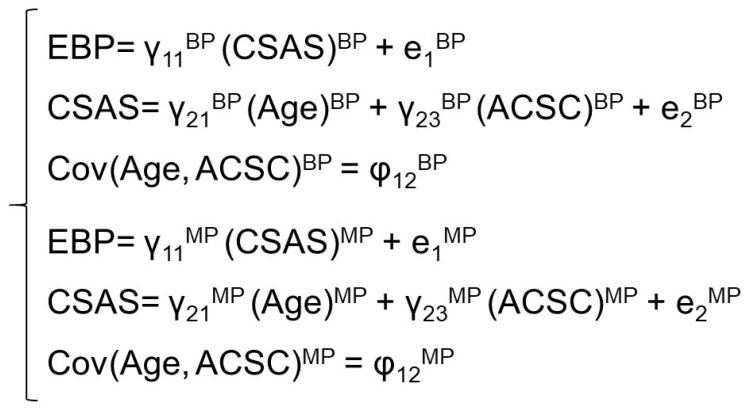
The formal form of the structural equation model. BP—bachelor’s programme, MP—master’s programme, EBP—attitude towards evidence-based nursing practice, CSAS—attitude towards learning communicative competences, ACSC—activity-centred spiritual care.

**Figure 3 ijerph-18-12591-f003:**
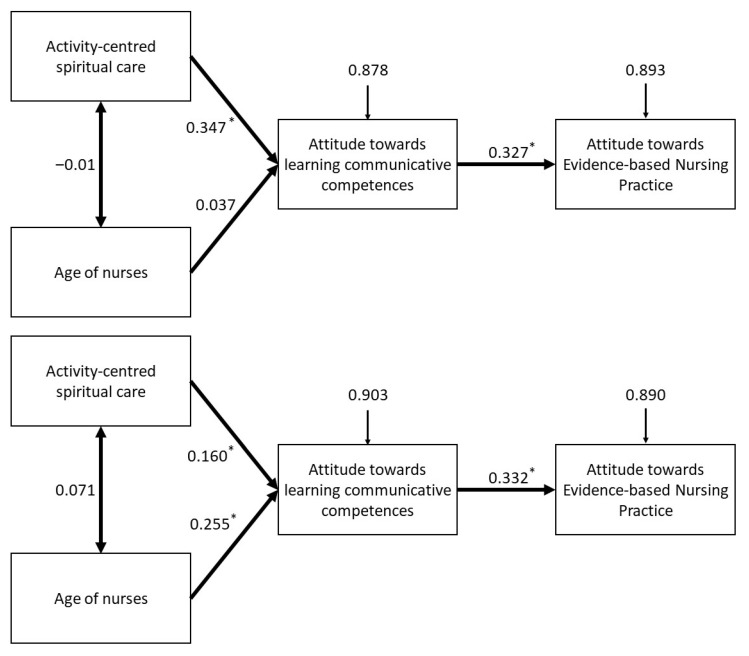
Pathway diagram for BP group (above) and MP group (below). Correlations between independent variables are represented with a double-sided arrow. Direct effects are represented with a one-sided arrow. The numbers above the arrow indicate the value of standardised regression weights. The numbers before the arrow show residual variances. The asterisk (*) mark stands for standard regression weights, meaning the *p* value is under 0.05.

**Table 1 ijerph-18-12591-t001:** Participant characteristics.

	Total (*n* = 1176)		Bachelor’s Programme (*n* = 459)		Master’s Programme (*n* = 717)		χ^2^	*p* Value *
	*n*	%	*n*	%	*n*	%		
Nursing School								
Medical University of Bialystok	162	13.8	16	3.4	147	20.5	158.4	<0.001
Medical University of Lublin	141	12.0	83	18.1	58	8.1		
Jagiellonian University Medical College	211	17.9	92	20.1	118	16.5		
Poznan University of Medical Sciences	171	14.5	70	15.2	101	14.1		
Medical University of Lodz	194	16.5	112	24.4	82	11.4		
Medical University of Silesia	79	6.7	0	0.0	79	11.0		
Medical University of Warsaw	219	18.6	87	18.9	132	18.4		
Gender								
Female	1104	93.9	425	92.6	679	94.7	2.163	0.141
Male	72	6.1	34	7.4	38	5.3		
Place of residence								
Countryside	309	26.3	190	41.3	119	16.7	100.4	<0.001
Village (population up to 50 thousand)	180	15.3	64	14.0	116	16.1		
Small town (51–200 thousand inhabitants)	144	12.3	33	7.2	111	15.5		
Large town (201–500 thousand inhabitants)	190	16.2	47	10.3	143	19.9		
City >500 thousand inhabitants	353	30.0	125	27.2	228	31.8		
Spiritual care training								
No	365	31.1	168	36.7	197	27.5	10.888	0.001
Yes	811	68.9	291	63.3	520	72.5		
EBP training								
No	1105	94.0	431	94.0	674	94.0	0.005	0.942
Yes	71	6.0	28	6.0	43	6.0		
Communication skills training								
No	960	81.6	374	81.4	586	81.7	0.012	0.915
Yes	216	18.4	85	18.6	131	18.3		

* Chi-squared test.

**Table 2 ijerph-18-12591-t002:** Comparison of two study groups in terms of variables.

Variable	Bachelor’s Programme		Master’s Programme		*t* _(df = 1174)_	*p* Value *	*D* ** (95% CI)
	M	SD	M	SD			
Attitude Towards Evidence-Based Nursing Practice	47.65	7.22	49.34	7.28	3.891	<0.001	0.23 (0.12; 0.35)
Attitude Towards Learning Communicative Competences	83.61	13.03	87.48	12.67	5.050	<0.001	0.30 (0.18; 0.42)
Activity-Centred Spiritual Care	33.71	9.15	34.79	8.47	2.064	0.039	0.12 (0.01; 0.24)

M—mean, SD—standard deviation, CI—confidence interval. * Student’s *t* test. ** Cohen’s *d* coefficient.

**Table 3 ijerph-18-12591-t003:** Standardised regression weights and test invariance across groups.

Construct/Hypothesis	Bachelor’s Programme				Master’s Programme				CMIN	*p* Value
	Estimate	SE	CR	*p* Value	Estimate	SE	CR	*p* Value		
ACSC → CSAS (H1)	0.347	0.041	8.47	0.000	0.160	0.035	4.56	0.000	9.643	0.002
CSAS → EBP (H2)	0.327	0.042	7.85	0.000	0.332	0.033	9.98	0.000	0.086	0.769
Age → CSAS (H3)	0.037	0.044	0.84	0.398	0.255	0.030	7.41	0.000	1.177	0.278
Age ↔ ACSC (H4)	−0.01	0.047	−0.21	0.834	0.071	0.037	1.91	0.057	3.260	0.071

SE—standard error, CR—critical ratio, EBP—attitude towards evidence-based nursing practice, CSAS—attitude towards learning communicative competences, ACSC—activity-centred spiritual care test invariance across groups, H—hypothesis.

## Data Availability

The data that support the findings of this study are available on request from the corresponding author (M.P.).

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
