# Peer review of "Perceiving the Role of Communication Skills as a Bridge between the Perception of Spiritual Care and Acceptance of Evidence-Based Nursing Practice—Empirical Model"

_ijerph, 2021, doi:10.3390/ijerph182312591_

Round 1
Reviewer 1 Report
Dear Authors,
Thank you very much for the results you present in this manuscript. It is an interesting and topical subject. However, some improvements are needed to increase the scientific rigour of the results presented:
You have designed and tested a predictive model using MPlus software. It would be necessary, at the end of the introduction, to better describe the hypotheses you wish to test, in terms of the model you are designing.
Reading the results you provide, I believe that the objective of your study can be improved, as you have finally obtained a predictive model.
Regarding the method, it is not clear why you use a sample of nursing students, together with a sample of postgraduate students; a priori, there may be numerous differences in the results that support your predictive model. Nor do they provide results as to whether there are differences between the two groups, which are heterogeneous.
It is also necessary to state whether the study has obtained permission from any Ethics Committee, and to insert the registration number of such permission.
For the questionnaires used, the alpha values obtained in their validation in the Polish population, as well as the interpretation of their scores (measurement ranges and cut-off points) should also be recorded.
The sampling procedure should be defined more comprehensively: it is understood that the sampling was convenience sampling, but this should be stated. Also, how the anonymity of the participants was guaranteed, if the questionnaires were collected together with the informed consent form, in which it is necessary to indicate personal data of the participant.
In the statistical analysis section, they should clarify what the critical values are for each of the parameters calculated to test the predictive model.
The discussion is generally well developed, although it would be interesting to include the impact or implications of your study in real practice, and to clarify why 2 heterogeneous study groups were used.
I encourage you to make these modifications to the manuscript, and to resubmit it for further revision.
Reviewer 2 Report
Through this study, I think the emphasis on the consideration of patients' spiritual care and the importance of communication skills for EBP was a very useful topic in the current nursing clinical field.
- Does it mean that there is no ban on The Local Personal Data Inspector in ethical considerations, replacing IRB?
- 2.4 Please present the reliability of the tool(Cronbach's Alpha) at the time of development and in this study in Variables
- It is not known what meaningful results the authors are presenting through the difference test in Table 1. participant characteristics. I think the technical description of the general characteristics of the subjects will be sufficient.
- For readers, statistical significance should be indicated by * on the value of the standard regression weight in Figure 3.
Round 2
Reviewer 1 Report
Dear authors,
Thank you very much for your improvements. Manuscript is now better rigorous. Congratulations.